# NSC828779 Alleviates Renal Tubulointerstitial Lesions Involving Interleukin-36 Signaling in Mice

**DOI:** 10.3390/cells10113060

**Published:** 2021-11-06

**Authors:** Shin-Ruen Yang, Szu-Chun Hung, Lichieh Julie Chu, Kuo-Feng Hua, Chyou-Wei Wei, I-Lin Tsai, Chih-Chin Kao, Chih-Chien Sung, Pauling Chu, Chung-Yao Wu, Ann Chen, Alexander T. H. Wu, Feng-Cheng Liu, Hsu-Shan Huang, Shuk-Man Ka

**Affiliations:** 1Department of Pathology, Tri-Service General Hospital, National Defense Medical Center, Taipei 11490, Taiwan; star2000yang@gmail.com (S.-R.Y.); ivan770401@gmail.com (C.-Y.W.); annchen31717@gmail.com (A.C.); 2Graduate Institute of Aerospace and Undersea Medicine, Department of Medicine, National Defense Medical Center, Taipei 11490, Taiwan; 3Division of Nephrology, Taipei Tzu Chi Hospital, Taipei 23142, Taiwan; szuchun.hung@gmail.com; 4Molecular Medicine Research Center, Chang Gung University, Taoyuan 33302, Taiwan; julie.chu.cgu@gmail.com; 5Liver Research Center, Chang Gung Memorial Hospital at Linkou, Gueishan, Taoyuan 33302, Taiwan; 6Department of Biotechnology and Animal Science, National Ilan University, Ilan 260007, Taiwan; kuofenghua@gmail.com; 7Department of Medical Research, China Medical University Hospital, China Medical University, Taichung 406040, Taiwan; 8Department of Nutrition, Master Program of Biomedical Nutrition, Hungkuang University, Taichung 433304, Taiwan; wcwnina@sunrise.hk.edu.tw; 9Department of Biochemistry and Molecular Cell Biology, School of Medicine, College of Medicine, Taipei Medical University, Taipei 11031, Taiwan; isabel10@tmu.edu.tw; 10Division of Nephrology, Department of Internal Medicine, Taipei Medical University Hospital, Taipei 11031, Taiwan; 121008@h.tmu.edu.tw; 11Division of Nephrology, Department of Internal Medicine, School of Medicine, College of Medicine, Taipei Medical University, Taipei 11031, Taiwan; 12Division of Nephrology, Department of Medicine, Tri-Service General Hospital, National Defense Medical Center, Taipei 11490, Taiwan; sungchihchien@gmail.com (C.-C.S.); pauling.chu@gmail.com (P.C.); 13The PhD Program for Translational Medicine, College of Medical Science and Technology, Taipei Medical University, Taipei 11031, Taiwan; chaw1211@tmu.edu.tw; 14Division of Rheumatology/Immunology and Allergy, Department of Medicine, Tri-Service General Hospital, National Defense Medical Center, Taipei 11490, Taiwan; lfc10399@gmail.com; 15Graduate Institute of Cancer Biology and Drug Discovery, College of Medical Science and Technology, Taipei Medical University, Taipei 11301, Taiwan; huanghs99@gmail.com; 16Graduate Institute of Medical Sciences, National Defense Medical Center, Taipei 11490, Taiwan

**Keywords:** salicylanilide derivative, tubulointerstitial lesions, IL-36α, mechanically induced constant pressure, IL-36α/NLRP3 inflammasome

## Abstract

Renal tubulointerstitial lesions (TILs), a common pathologic hallmark of chronic kidney disease that evolves to end-stage renal disease, is characterized by progressive inflammation and pronounced fibrosis of the kidney. However, current therapeutic approaches to treat these lesions remain largely ineffectual. Previously, we demonstrated that elevated IL-36α levels in human renal tissue and urine are implicated in impaired renal function, and IL-36 signaling enhances activation of NLRP3 inflammasome in a mouse model of TILs. Recently, we synthesized NSC828779, a salicylanilide derivative (protected by U.S. patents with US 8975255 B2 and US 9162993 B2), which inhibits activation of NF-κB signaling with high immunomodulatory potency and low IC_50_, and we hypothesized that it would be a potential drug candidate for renal TILs. The current study validated the therapeutic effects of NSC828779 on TILs using a mouse model of unilateral ureteral obstruction (UUO) and relevant cell models, including renal tubular epithelial cells under mechanically induced constant pressure. Treatment with NSC828779 improved renal lesions, as demonstrated by dramatically reduced severity of renal inflammation and fibrosis and decreased urinary cytokine levels in UUO mice. This small molecule specifically inhibits the IL-36α/NLRP3 inflammasome pathway. Based on these results, the beneficial outcome represents synergistic suppression of both the IL-36α-activated MAPK/NLRP3 inflammasome and STAT3- and Smad2/3-dependent fibrogenic signaling. NSC828779 appears justified as a new drug candidate to treat renal progressive inflammation and fibrosis.

## 1. Introduction 

Renal tubulointerstitial lesions (TILs), characterized by inflammation and fibrosis of the kidneys, are key pathological findings underlying progression to end-stage renal disease [1,2]. Unfortunately, current therapeutic approaches to these renal lesions remain clinically insufficient. IL-36α (IL-1F6), a proinflammatory cytokine, is essential for innate and adaptive immunity [3]. However, in psoriasis, it also triggers early activation of STAT3, NF-κB, and MAPKs in keratinocytes [4]. Recently, we showed that IL-36α facilitates activation of NLRP3 inflammasome in development and progression of renal failure [5]. We found that activation of IL-36 signaling is implicated in renal inflammation and fibrosis in a mouse unilateral ureteral obstruction (UUO) model. Other studies also confirm that pathogenesis of renal TILs in the UUO model involves NLRP3 inflammasome activation [6,7,8,9]. Activation of NLRP3 inflammasome involves a priming signal from pathogen-associated molecular patterns and an activation signal (e.g., ATP) generated by damaged cells to produce mature IL-1β and IL-18 [10,11]. MAPKs serve as one of the priming signals for NLRP3 inflammasome in renal inflammation and fibrosis [12,13]. In addition, STAT3 signaling has been implicated in renal fibrotic processes of UUO mice [14,15,16]. Our recent study confirms that STAT3 signaling and NLRP3 inflammasome work together as a major pathogenic pathway underlying TILs in UUO [17]. 

Very recently, we generated a series of new 5-(2′,4′-difluorophenyl)-salicylanilide derivatives and their ring-fused analogs, with anti-inflammatory properties, such as 2-hydroxy-*N*-[3-(trifluoromethyl)phenyl]benzamide [18], *N*-(4-chloro-2-fluorophenyl)-2-hydroxybenzamide [19], and *N*-(3-chloro-4-fluorophenyl)-2-hydroxybenzamide [20]. These small compounds inhibit the expression or activities of RANKL and RANKL-related effector genes, including NF-κB, nuclear factor of activated T cells, c-fas, triiodothyronine receptor auxiliary protein, and cathepsin K in RANKL-induced osteoclastogenesis [21,22]. During the course of making derivatives from synthetic analogs of 2-hydroxy-*N*-[3-(trifluoromethyl)phenyl]benzamide, we developed NSC828779, a salicylanilide derivative (U.S. patents with US 8975255 B2 and US 9162993 B2), which shows potent immunomodulatory properties [23]. In this study, we tested the hypothesis that NSC828779 has therapeutic effects on renal inflammation and fibrosis using a mouse model of UUO and relevant cell models, and further investigated the mode of action of this small molecule.

## 2. Materials and Methods

### 2.1. Synthesis of NSC828779

NSC828779, a salicylanilide derivative, is 6-(2,4-difluorophenyl)-3-(3-(trifluoromethyl)phenyl)-2*H*-benzo[e][1,3]oxazine-2,4(3*H*)-dione (our U.S. patents for this small molecule are: US 8975255 B2 and US 9162993 B2), as described previously [21].

### 2.2. UUO Animal Model and Treatment Experiments

Eight-week-old female C57BL/6 mice were subjected to a dorsolateral incision after proper anesthesia. Then the left ureter was permanently ligated as described previously [24]. Mice were then divided into two groups: (1) those treated with NSC828779 (daily dose of 10 mg/kg body weight, *i.p.*) dissolved in polyethylene glycol 400 (PEG 400) (Sigma-Aldrich, St. Louis, MO, USA) (UUO + NSC828779) one day after UUO induction, and (2) those that received vehicle (PEG 400) only, to serve as disease controls. Sham-operated mice, which received an identical surgical procedure, but without ureteric ligation (sham controls), and sham-operated mice treated with NSC828779 (sham + NSC828779) served as controls. Seven mice per group was used. At day 7 or 14, mice were euthanized, and their renal tissues and pelvic urine were collected [25]. All animal experiments were performed with approval of the Institutional Animal Care and Use Committee of the National Defense Medical Center, Taiwan, in compliance with the NIH Guide for the Care and Use of Laboratory Animals.

### 2.3. Histopathology and Immunohistochemistry

Renal tissues were fixed in 10% buffered formalin, embedded in paraffin, and cut into 3-μm slices. Hematoxylin and eosin or Masson trichrome staining were used for histologic examination under light microscopy. Renal TILs were scored in 40 consecutive selected fields at a magnification of ×400, as described previously [5]. For IHC analysis, renal tissue sections were stained with antibodies against F4/80 (Bio-Rad, Hercules, CA, USA), CD3 (Dako, Glostrup, Denmark), collagen (Col)-I, Col-III (Southern Biotech, Birmingham, AL, USA), p-Smad2/3 (Santa Cruz, CA, USA) or IL-36α (R&D Systems, Minneapolis, MN, USA). Quantitative analysis was conducted at a magnification of ×400 in 40 selected consecutive fields from both cortical and outer medullary regions using PAX-it image analysis software (Pax-it; Paxcam, Villa Park, IL, USA), as described previously [5].

### 2.4. Cultured Cells

A mouse renal tubular epithelial cell (TEC) line (CRL-2038) purchased from (Bioresource Collection and Research Center, Hsinchu, Taiwan) was maintained in F12/DMEM (Invitrogen, Waltham, MA, USA) with 5% FBS and 5 nM dexamethasone (Sigma-Aldrich, St. Louis, MO, USA). A mouse J774A.1 macrophage cell line (TIB-67) was purchased from the American Type Culture Collection (Manassas, VA, USA), and maintained in RPMI 1640 with 10% FBS, as described previously [5,17].

### 2.5. Western Blot Analysis 

Protein extraction from renal tissues and cultured cells was conducted. Each target protein was analyzed by SDS-PAGE and immunoblotting using respective antibodies against NLRP3, caspase-1 (Adipogen, San Diego, CA, USA), IL-1β, IL-36α (R&D Systems), p-STAT3, p-JNK, p-ERK, p-p38 (Cell Signaling, Danvers, MA, USA) or b-actin (Santa Cruz), as described previously [5].

### 2.6. ELISA

Levels of IL-1b (Cat# DY401) and MCP-1 (Cat# DY479) in urine and/or supernatants of cultured cells were determined by ELISA according to the manufacturer’s instructions (R&D Systems). 

### 2.7. Real-Time PCR Assay 

RNA was extracted with REzol (Protech Technology, Taipei, Taiwan) from renal tissues and cultured cells, and cDNA was prepared as described previously [5]. Real-time PCR analysis was conducted according to the manufacturer’s instructions. Primer pairs used for analysis are as follows: mouse IL-36α forward: 5′-CAGCATCACCTTCGCTTAGAC-3′; mouse IL-36α reverse: 5′-AGTGTCCAGATATTGGCATGG-3′; mouse NLRP3 forward: 5′ CTGTGTGTGGGACTGAAGCAC-3′; mouse NLRP3 reverse: 5′-GCAGC CCTGCTGTTTCAGCAC-3′; mouse IL-1β forward: 5′-CCAGGATGAGGACATGAGCACC-3′; mouse IL-1β reverse: 5′-TTCTCTGCAGACTCAAACTCCAC-3′; mouse caspase-1 forward: 5′- ACTGTACAACCGGAGTAATACGG-3′; mouse caspase-1 reverse: 5′- CACGGA AGGCCATGCCAGTGA-3′; and mouse GAPDH forward: 5′-TCCGCCCCTTCTGCCGATG-3′; mouse GAPDH reverse: 5′-CACGGAAGGCCATGCCAGTGA-3′.

### 2.8. Mechanically Induced Constant Pressure Model

To simulate the microenvironment in the renal pelvis of diseased kidneys of UUO mice, we employed a mechanically induced constant pressure (MICP) model in renal tubular epithelial cells, which we developed previously [26]. In brief, renal TECs were cultured in a closed system under 60 mmHg pressure maintained in a CO_2_ incubator for 24 h. Cell lysates were subjected to Western blot analysis for IL-36α production, as described above.

### 2.9. IL-36a-Mediated Activation of NLRP3 Inflammasome in Renal TECs and Macrophages

To detect the priming signal of NLRP3 inflammasome activation, TECs or J774A.1 macrophages were treated with or without NSC828779 for 30 min and then incubated with recombinant IL-36α (rIL-36α) (150 ng/mL) (BioLegend, San Diego, CA, USA) or LPS (0.5 mg/mL) (Sigma-Aldrich) for 5.5 h. For the second signal of inflammasome activation, after treatment with NSC828779 for 30 min, both types of cells were primed with rIL-36α for 24 h (TECs) or rIL-36α for 30 min and LPS 5.5 h (macrophages), and then ATP (Invitrogen) was added, as described previously [5]. Supernatant and cell lysates were subjected to ELISA or Western blot analysis according to the manufacturer’s instructions. 

### 2.10. Reporter Assay for NF-κB Activation

J774A.1 macrophages were stably transfected with an NF-κB-inducible reporter plasmid, pNiFty2-SEAP, to assess NF-κB transcriptional activity as described previously [27]. A total of 2 × 10_5_ cells per well in 0.5 mL of culture medium were added and allowed to grow overnight at 37 °C in a 5% CO_2_ incubator. Cells were then treated with NSC828779 (2.5, 5, 10 μM) for 2 h, followed by incubation with LPS for 24 h, as previously reported [27]. NF-κB transcriptional activity was determined with QUANTI-Blue medium [27].

### 2.11. In-Silico Molecular Docking Analyses 

The ligand, NSC828779, was subjected to molecular docking using Avogadro software [28], and then it was transformed to Protein Data Bank format using the PyMOL Molecular Graphics System, vers. 1.2.r3pre (Schrodinger, LLC, New York, NY, USA). The PDB file of the 3D structure of the receptor and the crystal structure of the STAT3 (PDB; 4ZIA) were retrieved from the Protein Data Bank. PDB file formats of the ligand NSC828779 and the receptor STAT3 were then converted to an AutoDock pdbqt format using AutoDock Vina (Vers, 0.8, The Scripps Research Institute, San Diego, CA, USA) [29]. Water molecules were removed for pre-docking preparation, followed by addition of Kolmman charges and hydrogen atoms. Binding energy values (kcal/mol) were expressed according to hydrogen bonds, electrostatic, and hydrophobic interactions of the optimal orientation of the ligand-receptor complex. Three-dimensional graphical representations of the ligand–receptor complex determined with PyMOL software to analyze H-bond interactions, interacting amino acid residues, binding atoms, binding affinities on the ligand and receptor were used with Discovery Studio Visualizer, vers. 19.1.0.18287 (BIOVIA, San Diego, CA, USA) to produce a two-dimensional graphical illustration of ligand-receptor interactions [30].

### 2.12. Statistical Analysis

Data are presented as means ± SEM. Comparisons between the two groups were conducted with Student’s *t*-test for animal experiments. In vitro data were analyzed using one-way ANOVA, followed by Scheffe’s test. A *p* value <  0.05 was considered statistically significant.

## 3. Results

### 3.1. NSC828779 Improves Renal TILs in UUO Mice

This study employed both early stage (one week after disease induction) and later stage (two weeks after disease induction) mouse UUO models to validate beneficial effects of NSC828779, a novel benzamide-linked small molecule, on TILs, followed by a series of experiments on the mechanism of action of the new compound using renal TECs and macrophages. Besides, the NSC828779 treatment showed no detectable systemic side effects in the mice.

### 3.2. Decreased Urine Cytokine Levels

Urine samples were collected from dilated renal pelves. Although increased urine levels of IL-1b and MCP-1 were observed at day 7 in UUO mice treated with vehicle only (UUO + Vehicle), followed by a steep increase at day 14 (Figure 1A,B), these effects of proinflammatory cytokines in urine were significantly inhibited in UUO + NSC828779 mice at both days 7 and 14; however, there were higher urine levels of these proinflammatory cytokines at day 14 than those of day 7. There was no significant difference in urine levels of these proteins between sham + NSC828779 mice and sham control mice.

### 3.3. Alleviated Pathological Changes

Under light microscopy, UUO + NSC828779 mice showed greatly reduced severity of renal lesions, including glomerular collapse, and mononuclear leukocyte infiltration and fibrotic changes in the renal interstitium at days 7 and 14, compared with UUO + Vehicle mice (Figure 2A–D); however, there were higher levels of fibrotic changes at day 14 than those of day 7. There was no significant difference in urine levels of these proteins between sham + NSC828779 mice and sham control mice. Furthermore, although UUO + Vehicle mice showed diffuse infiltration of F4/80^+^ monocytes/macrophages and CD3^+^ T cells in renal interstitial tissues on day 7, which continued to increase until day 14, this inflammatory cell infiltration was significantly suppressed in UUO + NSC828779 mice (Figure 2E–H). Again, significantly reduced fibrosis of the renal cortex was observed in UUO + NSC828779 mice, as demonstrated by decreased expression of Col-I (Appendix A), Col-III (Figure 2I,J) and p-Smad2/3 (Figure 2K,L) in the kidney; however, there were higher levels of Col-III at day 14 than those of day 7. 

As shown in Figure 3, enhanced IL-36α expression was seen in renal TECs of UUO + Vehicle mice, as demonstrated by IHC, but this effect was greatly inhibited in UUO + NSC828779 mice on days 7 and 14 (Figure 3A,B), after disease induction.

### 3.4. NSC828779 Inhibits the IL-36α/NLRP3 Inflammasome Pathway 

#### 3.4.1. Renal IL-36α Expression and NLRP3 Inflammasome Activation in UUO Mice

In the present study, significantly reduced mRNA levels of IL-36α were noted in UUO + NSC828779 mice (Figure 3C), compared with those of UUO + Vehicle mice at days 7 and 14. Furthermore, assessment of activation of NLRP3 inflammasome in renal tissues revealed that treatment with NSC828779 significantly decreased the expression of renal mRNA of NLRP3, IL-1β, and caspase-1 in UUO + NSC828779 mice compared with UUO + Vehicle mice, which showed increased mRNA levels of these proteins compared with sham-operated control mice (Figure 3D–F); however, there were higher mRNA levels of NLRP3, IL-1β, and caspase-1 at day 14 than those of day 7. Similarly, UUO + Vehicle mice exhibited increased protein expression of renal IL-36α, NLRP3, active caspase-1 and IL-1β compared with control mice, whereas these increases were inhibited in UUO + NSC828779 mice at both days 7 and 14 after disease induction (Figure 3G,H); however, there was a higher protein level of IL-1β at day 14 than those of day 7.

#### 3.4.2. IL-36α Expression Mediated by a Cell-Based, MICP Model or H_2_O_2_ in Renal TECs

Increased renal expression of IL-36α has been observed in the UUO model [5,31]. Although activation levels of IL-36α in renal TECs under MICP and H_2_O_2_, treated with saline only, increased compared with those of controls, these increases were inhibited in NSC828779-treated renal TECs (Figure 4A,B). In the present study, we found that NLRP3 expression levels in rIL-36α-stimulated renal TECs were significantly reduced by NSC828779 treatment (Figure 4C). In addition, we showed that NSC828779 inhibited ATP-induced IL-1β secretion and caspase-1 activation in rIL-36α-primed renal TECs (Figure 4D). These results suggest that NSC828779 reduces rIL-36α-meditated NLRP3 inflammasome activation in renal TECs. 

#### 3.4.3. NSC828779 Inhibits NLRP3 Inflammasome Activation in IL-36α-Primed Macrophages

We examined the effect of NSC828779 on rIL-36α-induced NLRP3 inflammasome activation in macrophages. The results showed that NSC828779 significantly reduced protein levels of mature IL-1β and caspase-1 in rIL-36α-stimulated macrophages, as demonstrated by ELISA (Figure 5A) and Western blot analysis (Figure 5B). Next, we investigated potential molecular mechanisms underlying NSC828779 suppression of IL-36-mediated NLRP3 inflammasome activation. With NSC828779 treatment, rIL-36α-stimulated macrophages showed significantly decreased protein and mRNA levels of NLRP3 (Figure 5C,D). We further examined whether NSC828779 could regulate NF-kB and MAPKs, which are implicated in NLRP3 inflammasome activation in rIL-36α-stimulated macrophages. Significantly reduced phosphorylation of IkB, JNK1/2, ERK1/2, and p38 was observed in rIL-36α-stimulated macrophages treated with NSC828779 (Figure 5E). Moreover, inhibition of ERK1/2 and JNK1/2 by PD98059 and SP600125, respectively, significantly inhibited NLRP3 expression, although inhibition of p38 by SB203580 failed to affect NLRP3 expression in IL-36α-stimulated macrophages (Figure 5F). These results suggest that NSC828779 treatment inhibited IL-36α-mediated NLRP3 inflammasome activation through ERK1/2- and JNK1/2-dependent pathways in macrophages. In parallel, we tested whether NSC828779 could inhibit NLRP3 inflammasome activation in LPS-primed macrophages. LPS-primed, ATP-activated macrophages secreted significantly higher IL-1β levels than saline controls, but treatment with NSC828779 abolished this effect in a dose-dependent manner, as documented using ELISA (Figure 6A). Consistent with this effect, it was clear that treatment with NSC828779 decreased production of IL-1β and caspase-1, as demonstrated in Western blot analysis (Figure 6B). NLRP3 and pro-IL-1β expression levels in LPS-primed macrophages were significantly decreased by NSC828779 treatment (Figure 6C). Moreover, NSC828779 reduced NLRP3 expression in LPS-primed macrophages and these effects involved suppressed NF-κB activation (Figure 6D). Next, we examined whether NSC828779 could influence K^+^ efflux, an upstream signaling event for NLRP3 activation [32]. Treatment with NSC828779 inhibited IL-1β secretion (Figure 6E) and production of IL-1β (Figure 6F) in LPS-primed macrophages with K^+^ free medium. Collectively, these results suggest that NSC828779 treatment reduced IL-1β production by inhibiting the NF-κB/NLRP3 inflammasome axis in LPS-primed macrophages.

### 3.5. NSC828779 Inhibits STAT3 Signaling

Renal tissues of UUO mice were subjected to Western blot analysis to determine expression levels of p-STAT3. Although increased p-STAT3 activation was seen in UUO + Vehicle mice, this effect was suppressed in UUO + NSC828779 mice (Figure 7A,B); there was a higher level of p-STAT3 at day 14 than that of day 7. Treatment with NSC828779 reduced the phosphorylation level of STAT3 in rIL-36α-primed macrophages (Figure 7C). In parallel, an in-silico molecular docking analysis revealed that NSC828779 docked well with the STAT3 binding cavity, displaying stronger binding affinity (−9.2 kcal/mol), more robust interactions, and shorter interaction distances (2.63~4.07 Å) than the interaction observed in STAT3 and a standard inhibitor, SH-4-54 (−8.4 kcal/mol, 2.03~3.61 Å). The NSC828779-STAT3 complex involves H-bonds with SER-113, a carbon-hydrogen bond between TRP-43 and SER-48, halogen bonding with ALA-44, ALA-47, ALA-106, GLN-41, and TRP-110, including a π-stacked interaction with TRP-110 in the receptor-binding pocket (Figure 7D) (Appendix A). These findings suggest that NSC828779 may be an inhibitor of the STAT3 signaling.

In the present study, we showed that NSC828779 may be a new drug candidate for treating TILs, as demonstrated by its dramatic reduction of the severity of renal inflammation and fibrosis in UUO mice. Beneficial outcomes with this small molecule may represent a synergistic effect of combined suppression of IL-36α-activated MAPK/NLRP3 inflammasome and the STAT3- and Smad2/3-dependent fibrogenic pathway (Figure 8).

Previously, we [5] and others [31] showed that IL-36α is highly upregulated in kidneys of UUO mice. A variety of epithelial cells express IL-36 cytokines, involved in activating NF-κB/MAPK pathways and promoting inflammatory responses in respiratory diseases, arthritis, renal disease, or dermatitis [33,34,35]. We also demonstrated that IL-36 signaling facilitates activation of NLRP3 inflammasome, thereby increasing the production of IL-1b, in TILs, which represent a typical prognostic feature for renal inflammation and fibrosis [5]. Therefore, NSC828779, which inhibits NF-κB signaling, may exert therapeutic effects on TILs by inhibiting IL-36 signaling-mediated activation of NLRP3 inflammasome in renal lesions. Furthermore, NLRP3 inflammasome are implicated in renal inflammation and fibrosis of UUO mice [17]. In the present study, NSC828779 inhibited activation of NLRP3 inflammasome, as demonstrated by greatly reduced renal levels of NLRP3, caspase-1, IL-1β protein, and urinary cytokines, including IL-1β. Again, these results suggest that inhibition of NLRP3 inflammasome underlies therapeutic effects of NSC828779 on renal TILs in UUO mice. Inhibition of NLRP3 inflammasome in NSC828779-treated renal TECs under an MICP model confirms the therapeutic effects of NSC828779 in this mouse UUO model through negative regulation of both priming and activation signals of NLRP3 inflammasome in renal TECs, the major cell type involved in the pathogenesis of renal TILs of UUO mice.

Recently, we also showed that blockade of the ERK1/2, JNK1/2, and p38 signaling pathways inhibits IL-1β expression in activated macrophages. In the present study, NSC828779 inhibited phosphorylation of p38 MAPK, JNK1/2, and ERK1/2 in rIL-36α-activated macrophages. Blockade of ERK1/2 or JNK1/2 inhibited NLRP3 expression, while inhibition of p38 by SB203580 did not affect NLRP3 expression in rIL-36α-primed macrophages (Figure 5F). Therefore, these findings suggest that NSC828779 may inhibit IL-36α-mediated NLRP3 inflammasome activation via the ERK1/2- and JNK1/2-dependent signaling pathways. In the present study, we evaluated the inhibitory effects of NSC828779 on phosphorylation of p38, JNK, and ERK by using specific inhibitors, respectively, although we did not detect the total form of those kinases (Figure 5F). However, this issue is worth further investigation for the mechanism of action of the small molecule.

MAPKs pathway-dependent phosphorylation of Smad2/3 is essential for activation of TGF-β/Smad signaling in development of fibrosis in various diseases [36,37]. In the present study, we showed that treatment with NSC828779 decreased fibrous tissue production and collagen accumulation in kidneys of UUO mice, accompanied by reduced renal expression of p-Smad2/3 protein. In this regard, TGF-β exerts its biological effects by activating its downstream intracellular substrates, Smad2 and Smad3 [38,39,40]. Blockade of downstream targets of the TGF-β/Smad signaling pathway inhibits both renal fibrosis and inflammation by blocking both the Smad2/3 and NF-κB signaling pathways in rat remnant kidney and UUO models [41,42,43]. Collectively, we proposed that TGF-β/Smad signaling may be involved in the mechanism of action of NSC828779 through crosstalk with the IL-36/MAPK axis.

Depletion of STAT3 inhibits IL-36– and IL-17A–mediated induction of IκBζ [4]. Furthermore, increased expression of active STAT3 is often detectable in the epidermis of psoriatic lesions, and pharmacological intervention to inhibit STAT3 improved psoriasis-like skin lesions in mice [44]. Suppression of STAT3 in renal TECs [45] or macrophages [46] inhibits renal fibrosis. The present study shows that NSC828779 inhibited renal p-STAT3 levels in UUO mice and p-STAT3 levels in rIL-36α-primed macrophages, suggesting that the inhibitory effect rendered by NSC828779 on IL-36α-activated STAT3 signaling is partly through blockade of STAT3 phosphorylation. Furthermore, we [17] and others [15,16] demonstrated that STAT3 signaling is involved in the development of TILs in UUO mice, in association with NF-κB-mediated inflammatory responses [47,48], and TGF-β1-dependent phosphorylation of Smad3 that promotes STAT3 activation in injured kidneys [49,50]. Altogether, our results confirm that inhibition of STAT3 activation could also be a mechanism of action for NSC828779 in TILs in the mouse UUO model.

In the present study, clinical relevance of urine protein levels of IL-1β and MCP1 in CKD and UUO model might be unclear, and to further detect serum levels of IL-1β and MCP1 might be useful. Besides, to assess the effects of the NSC828779 on kidney functions as demonstrated by levels of serum creatinine and blood urea detection in different groups might be helpful.

In the present study, a comparison between sham controls and those treated with the compound were made and there were no detectable systemic side effects in the mice that received NSC828779, also justifying it as a highly potential drug candidate. Besides, mitochondrial alterations and redox damage occurred after 14 days UUO in the rat kidney [51]. This issue is worth further investigation for the mechanism of action of the small molecule.

## 4. Conclusions

In summary, this study successfully validated NSC828779 as a promising drug candidate for treating renal TILs. This compound was effective in early or developmental stages of renal lesions by blocking IL-36α, which facilitates NLRP3 inflammasome and the TGF-β/STAT3/Smad2/3-mediated fibrogenic pathway, probably the major mechanism of action of the compound. Moreover, NSC828779 may be a suitable drug candidate to prevent progression of chronic kidney disease to end-stage renal disease.

## Figures and Tables

**Figure 1 cells-10-03060-f001:**
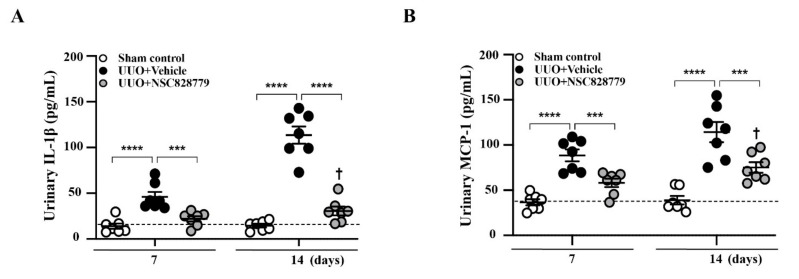
NSC828779 reduces urine levels of proinflammatory cytokines. ELISA analysis of urine protein levels of (**A**) IL-1β, and (**B**) MCP-1 collected from the dilated pelvis. The horizonal dashed line indicates the mean for sham control + NSC828779. Data are shown as means ± SEM of seven mice per group. NSC828779, a salicylanilide derivative. UUO, unilateral ureter obstruction. *** *p* < 0.005, **** *p* < 0.001. † *p* < 0.05 Day 7 versus Day14 of UUO + NSC828779.

**Figure 2 cells-10-03060-f002:**
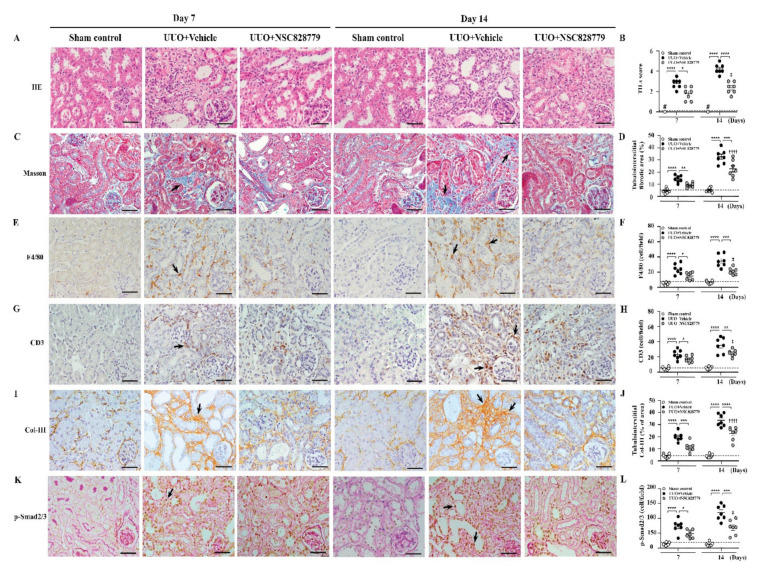
NSC828779 alleviates pathological changes in UUO mice. (**A**,**B**) H&E staining and scoring. (**C**,**D**) Masson trichrome staining and scoring of tubulointerstitial fibrotic area. IHC and scoring for F4/80^+^ (**E**,**F**), CD3^+^ (**G**,**H**), Col-III (**I**,**J**) and p-Smad2/3 (**K**,**L**). Original magnification, each 400×. Data are shown as means ± SEM of seven mice per group. The bar indicates 20 µm. Arrows indicate positive staining. The horizonal dashed line indicates the mean for sham control + NSC828779. NSC828779, a salicylanilide derivative. UUO, unilateral ureter obstruction; TILs, tubulointerstitial lesions. H&E staining, hematoxylin and eosin stain. Col-III, collagen III. # not detectable. * *p <* 0.05, ** *p <* 0.01, *** *p* < 0.005, **** *p* < 0.001, †††† *p* < 0.001 Day 7 versus Day14 of UUO + NSC828779, ‡ no significant difference Day 7 versus Day14 of UUO + NSC828779.

**Figure 3 cells-10-03060-f003:**
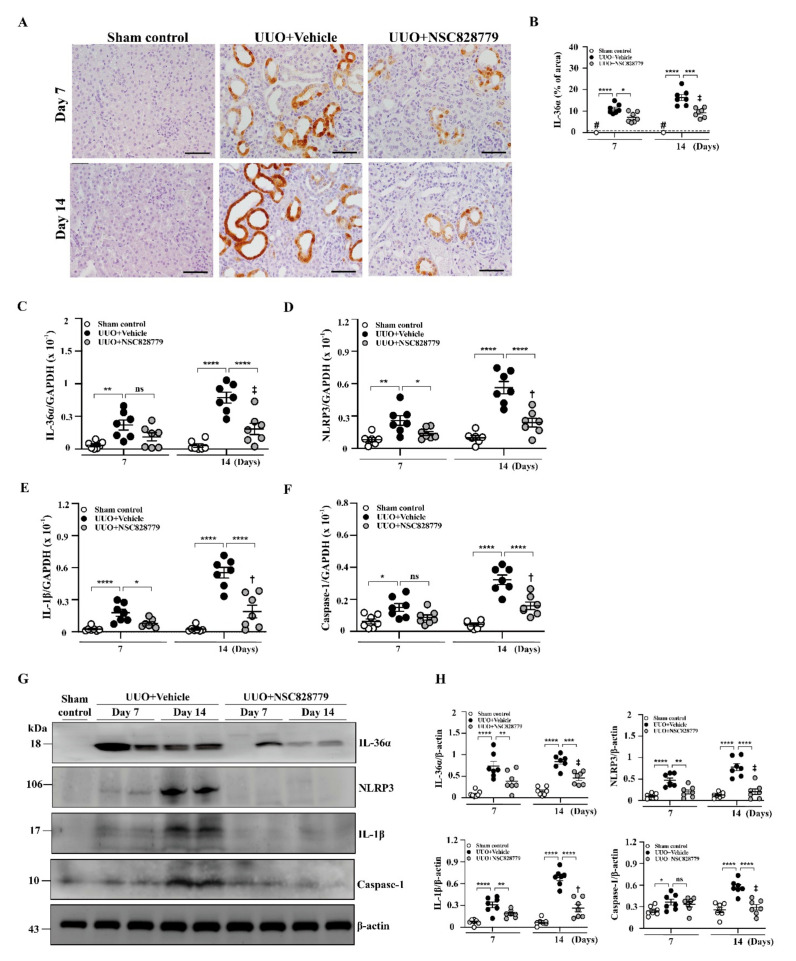
Renal IL-36α expression and NLRP3 inflammasome activation in UUO mice. (**A**,**B**) IHC and scoring for IL-36α. Original magnification, each 400×. Real-time PCR assay for mRNA levels of renal IL-36α (**C**), NLRP3 (**D**), IL-1β (**E**), and caspasae-1 (**F**). Western blot analysis for renal (**G**) IL-36α, NLRP3, IL-1β, and caspase-1. (**H**) Semi-quantitative analysis. Data are shown as means ± SEM of seven mice per group. The horizonal dashed line indicates the mean for sham control + NSC828779. NSC828779, a salicylanilide derivative. UUO, unilateral ureter obstruction. #not detectable. ns, no significant difference. * *p <* 0.05, ** *p <* 0.01, *** *p* < 0.005, **** *p* < 0.001. † *p*< 0.05 Day 7 versus Day14 of UUO + NSC828779, ‡ no significant difference Day 7 versus Day14 of UUO + NSC828779.

**Figure 4 cells-10-03060-f004:**
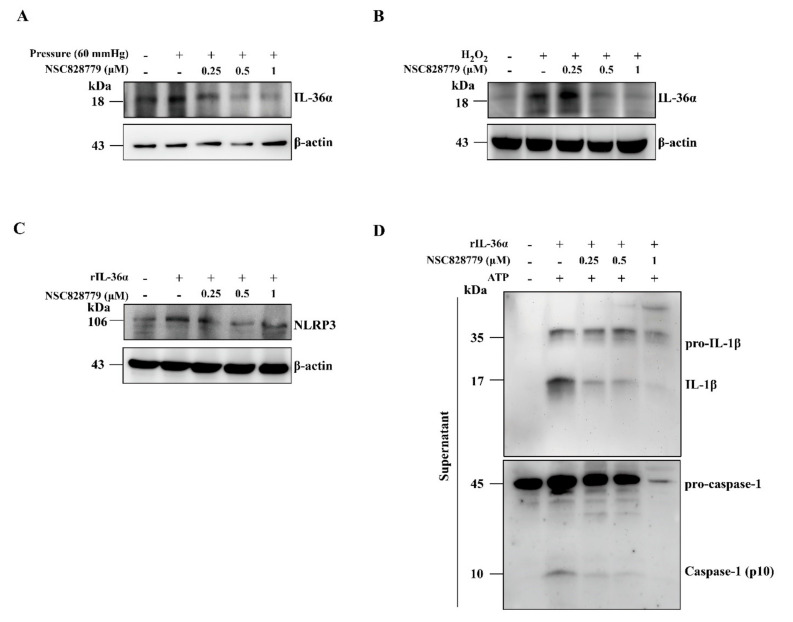
NSC828779 inhibits IL-36a expression in renal TECs induced by a cell-based, mechanically induced constant pressure model or H_2_O_2_. Renal TECs (a murine renal tubular epithelial cell line) were incubated for 30 min with or without NSC828779, cultured in a MICP model at 60 mmHg or H_2_O_2_ for 24 h, with or without 10 mM ATP (1 h). Western blot analysis for IL-36α protein expression in cells by MICP (**A**) and H_2_O_2_ (**B**). Cells were incubated for 30 min with or without NSC828779 and 24 h with or without 300 ng/mL of rIL-36α for NLRP3 expression (**C**). Representative Western blots for IL-1b and caspase-1 (**D**) in the supernatant of cells incubated for 30 min with or without NSC828779, then for 24 h with or without 300 ng/mL of rIL-36α, and with or without ATP (1 h). Data are shown as means ± SEM for three separate experiments, and each experiment was performed in triplicate. NSC828779, a salicylanilide derivative. TECs, tubular epithelial cells. MICP, mechanically induced constant pressure.

**Figure 5 cells-10-03060-f005:**
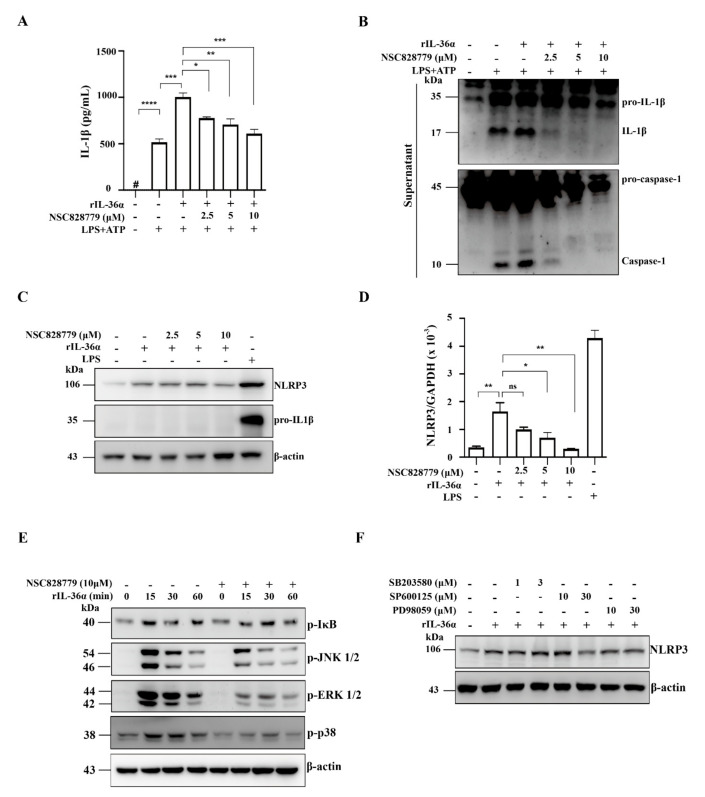
NSC828779 inhibits NLRP3 inflammasome activation in IL-36a-primed macrophages. J774A.1 macrophages were incubated for 30 min with or without NSC828779, then for 0.5 h rIL-36α with 0.5 µg/mL of LPS, with or without 5 mM ATP (30 min) for IL-1β levels (**A**) in culture media measured using ELISA or (**B**) IL-1β, caspase-1 by Western blot analysis. Cells were incubated for 30 min with or without NSC828779 and 6 h with or without 150 ng/mL of IL-36α for Western blots of (**C**) NLRP3 and pro-IL-1b or real-time PCR for mRNA levels of NLRP3 (**D**). Cells were incubated with for 30 min with or without NSC828779, then with rIL-36α for 15, 30, or 60 min. (**E**) Phosphorylated IkB, ERK1/2, JNK1/2, and p38 in cell lysates, measured by Western blot analysis. (F) Levels of NLRP3 in cell lysates by Western blot analysis. J774A.1 macrophages were incubated for 0.5 h with PD SP or SB, followed by incubation for 5.5 h with IL-36α at 150 ng/mL. Data represent three separate experiments. Data are shown as means ± SEM for three separate experiments, and each experiment was performed in triplicate. NSC828779, a salicylanilide derivative. PD98059, an inhibitor of p-ERK1/2. SB203580, an inhibitor of p-JNK1/2. SP600125, an inhibitor of p-p38. # not detectable. ns, no significant difference. * *p <* 0.05, ** *p <* 0.01, *** *p* < 0.005, **** *p* < 0.001.

**Figure 6 cells-10-03060-f006:**
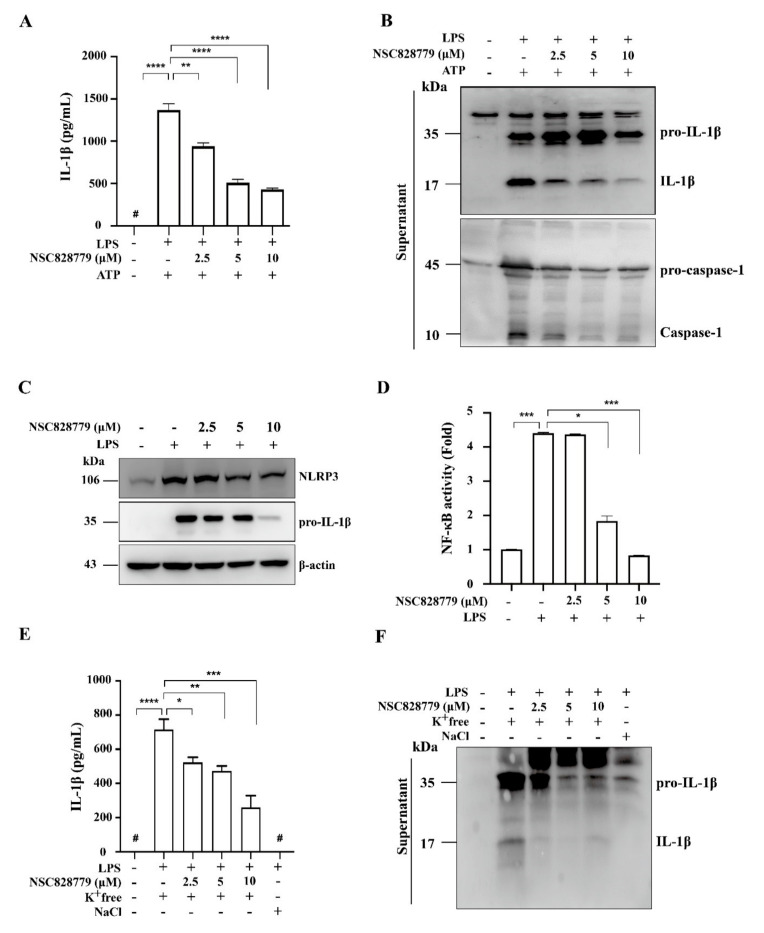
NSC828779 inhibits NLRP3 inflammasome in LPS-primed macrophages. J774A.1 macrophages were incubated for 30 min with or without NSC828779, then for 5.5 h with or without 1 µg/mL of LPS, with or without 5 mM ATP (30 min) for IL-1β levels (**A**) in culture media measured using ELISA or IL-1β and caspase-1 (**B**), determined by Western blot analysis. IL-1β and caspase-1 release into culture medium was assayed in extracted supernatants. Please (**C**) Levels of NLRP3 and pro-IL-1β expression levels in cell lysates by Western blot analysis. J-Blue cells were incubated for 30 min with or without NSC828779 and 24 h with or without 1 µg/mL of LPS for activation levels of NF-κB (**D**) determined using an NF-κB reporter assay. IL-1β expression measured by ELISA (**E**) and Western blot analysis (**F**) in J774A.1 macrophages incubated for 5 h with LPS followed by incubation for 0.5 h with NSC828779 followed by K^+^ free medium and NaCl medium for 2 h. Data are shown as means ± SEM for three separate experiments, and each experiment was performed in triplicate. NSC828779, a salicylanilide derivative. SEAP, secreted alkaline phosphatase. # not detectable. * *p <* 0.05, ** *p <* 0.01, *** *p <* 0.005, **** *p <* 0.001.

**Figure 7 cells-10-03060-f007:**
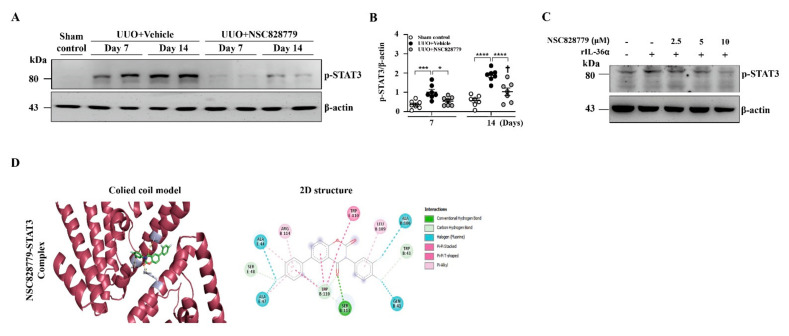
NSC828779 inhibits STAT3 activation and comparative docking profiles. (**A**) p-STAT3 in renal tissues by Western blot analysis. (**B**) Semi-quantitative analysis. Data are shown as means ± SEM of seven mice per group. J774A.1 macrophages were incubated for 30 min with or without NSC828779, then for 1 h with or without 150 ng/mL of IL-36α. (**C**) p-STAT3 by Western blot analysis. (**D**) In-silico molecular docking. 3D and 2D representation of ligand-receptor interactions of NSC828779 with STAT3 in the receptor-binding pocket. NSC828779, a salicylanilide derivative. UUO, unilateral ureter obstruction. * *p <* 0.05, *** *p <* 0.005, **** *p <* 0.001. † *p* < 0.05 Day 7 versus Day14 of UUO + NSC828779, ‡ no significant difference Day 7 versus Day14 of UUO + NSC828779.

**Figure 8 cells-10-03060-f008:**
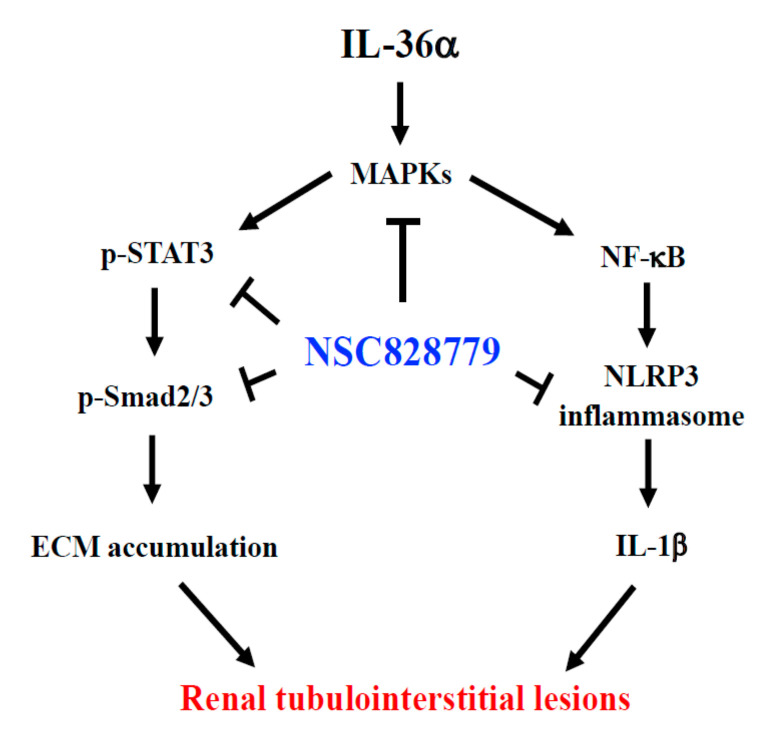
Schematic mechanism of action for therapeutic effects of NSC828779 on renal TILs. Treatment with NSC828779 improves renal inflammation and fibrosis in UUO model through a mechanism of action involving the IL-36α-activated MAPK/NLRP3 inflammasome and STAT3- and Smad2/3-dependent fibrogenic signaling. NSC828779, a salicylanilide derivative. UUO, unilateral ureter obstruction TILs, tubulointerstitial lesions. ECM, extracellular matrix.

## Data Availability

Data supporting the findings of this manuscript would be available from the corresponding author upon reasonable request.

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
