# Peer review of "NSC828779 Alleviates Renal Tubulointerstitial Lesions Involving Interleukin-36 Signaling in Mice"

_cells, 2021, doi:10.3390/cells10113060_

Round 1

Reviewer 1 Report

The Authors demonstrated that NSC828779 compound was effective to limit fibrosis and inflammation in mice after UUO and in vitro cells (Tubular epithelial cells and macrophages).

In my opinion, some minor changes are requested before full acceptance as follows:

1.The authors adopted a single dose of the drug even if there are different stages of fibrosis, at 7 days after UUO and at 14 days. I wonder why they did not used a higher dose to reduce the higher damage. Are there any side effects of NSC828779?

  1. Histological data might be better discussed because fibrosis and inflammation differently hit the renal cortex, containing glomeruli and the medulla.

3.Mitochondrial alterations and redox damage occurred after 14 days UUO in the rat kidney see Prieto-Carrasco et al Biology 10, 671, 2021. Why did not the Authors analyse mitochondrial status and eventually antioxidant restoration after NSC828779?

Author Response

Reviewer #1:
The Authors demonstrated that NSC828779 compound was effective to limit fibrosis and inflammation in mice after UUO and in vitro cells (Tubular epithelial cells and macrophages).
In my opinion, some minor changes are requested before full acceptance as follows:
Point 1: The authors adopted a single dose of the drug even if there are different stages of fibrosis, at 7 days after UUO and at 14 days. I wonder why they did not used a higher dose to reduce the higher damage. Are there any side effects of NSC828779?
Ans: We thank the reviewer very much for the critical comments and suggestions. In our pilot study, the dose used throughout our study, 10mg/kg/day was an optimal dose for the treatment of choice. In compliance with the important comment by the reviewer, we added the following to our revised version: “Besides, the NSC828779 treatment showed no detectable systemic side effects in the mice” in the Results section of the revised version (Lines 255-256, Page 11).
Point 2: Histological data might be better discussed because fibrosis and inflammation differently hit the renal cortex, containing glomeruli and the medulla.
Ans: We thank the reviewer very much for the excellent comments and suggestions. As kindly suggested, we revised the results for histological data by “Under light microscopy, UUO+NSC828779 mice showed greatly reduced severity of renal lesions, including glomerular collapse, and mononuclear leukocyte infiltration and fibrotic changes in the renal interstitium at days 7 and 14, compared with UUO+Vehicle mice (Figure 2A-D). …….” (Lines 267-269, Page 11) of the revised version.
Point 3: Mitochondrial alterations and redox damage occurred after 14 days UUO in the rat kidney see Prieto-Carrasco et al Biology 10, 671, 2021. Why did not the Authors analyse mitochondrial status and eventually antioxidant restoration after NSC828779?
Ans: We thank the reviewer very much for the excellent comments and suggestions. As kindly suggested, we added this important comment to the Discussion section of our revised version by “Besides, mitochondrial alterations and redox damage occurred after 14 days UUO in the rat kidney (Prieto-Carrasco et al Biology 10, 671, 2021). This issue is worth further investigation for the mechanism of action of the small molecule. (Lines 463-464, Page 18)

Reviewer 2 Report

I read with interest the manuscript entitled “NSC828779 alleviates renal tubulointerstitial lesions involving interleukin-36 signaling in mice· by Shin-Ruen Yang, et al that is intended to be published in Cells as an original manuscript.

No concerns with the study. It is in line with studies performed by the group. Hypothesis, development, and presentation of results are elegant, concise and reliable. New drugs potentially effective in prevention/treatment of renal chronic disease are very welcome.

Author Response

Reviewer #2:
I read with interest the manuscript entitled “NSC828779 alleviates renal tubulointerstitial lesions involving interleukin-36 signaling in mice· by Shin-Ruen Yang, et al that is intended to be published in Cells as an original manuscript.
No concerns with the study. It is in line with studies performed by the group. Hypothesis, development, and presentation of results are elegant, concise and reliable. New drugs potentially effective in prevention/treatment of renal chronic disease are very welcome.
Ans: We thank the reviewer for the comments very much!

Reviewer 3 Report

The manuscript by Yang et al et al. validated the therapeutic effects of NSC828779 in tubulointerstitial lesions (TILs) using well established unilateral ureteral obstruction (UUO) mouse model. There are concerns about experimental design, data analysis and interpretation. Poor data quality dampens the credibility of the manuscript.

Specific comments are listed below:

  1. Clinical relevance of urine protein levels of IL-1β and MCP1 in CKD and UUO model is unclear to this reviewer. Maybe serum levels of IL-1β and MCP1 might be useful. To assess the effects of NSC828779 on kidney functions- level of serum creatinine and blood urea detection in different groups are needed for this manuscript.
  2. Figure 2 is not convincing that NSC828779 reduce renal inflammation and fibrosis at day 7 and day 14. Even increases CD3 at day 14 (Fig G). Representing UUO HE and IHC images at day 7 and day 14 are not matching with the scoring data (Figs, CD EF, GH and KL). In some cases, pathological changes (tubular dilatation) are greater in sham group than other groups (Fig C). Inclusion of low magnification of HE and IHC images, complementary protein (Western blot) and mRNA level of fibrosis markers will improve the quality of the manuscript.
  3. What are the numbers are in scoring graphs in figure 2 and 3B? Are indicating area or mice? Not clear.
  4. Figures 4, 5 and 6 are representing from how many separate experiments..... are also not indicated in the manuscript.
  5. Is there any effect of NSC828779 on total p38, JNK and ERK?

Author Response

Reviewer #3:
The manuscript by Yang et al et al. validated the therapeutic effects of NSC828779 in tubulointerstitial lesions (TILs) using well established unilateral ureteral obstruction (UUO) mouse model. There are concerns about experimental design, data analysis and interpretation. Poor data quality dampens the credibility of the manuscript.
Specific comments are listed below:
Point 1: Clinical relevance of urine protein levels of IL-1β and MCP1 in CKD and UUO model is unclear to this reviewer. Maybe serum levels of IL-1β and MCP1 might be useful. To assess the effects of NSC828779 on kidney functions- level of serum creatinine and blood urea detection in different groups are needed for this manuscript.
Ans: We thank the reviewer very much for the critical comments and suggestions.
As kindly advised and suggested, we added the statements to the Discussion section of the revised version as follows:“In the present study, clinical relevance of urine protein levels of IL-1β and MCP1 in CKD and UUO model might be unclear, and to further detect serum levels of IL-1β and MCP1 might be useful. Besides, to assess the effects of the NSC828779 on kidney functions as demonstrated by levels of serum creatinine and blood urea detection in different groups might be helpful.(Line 455-459, Page 18).
Point 2: Figure 2 is not convincing that NSC828779 reduce renal inflammation and fibrosis at day 7 and day 14. Even increases CD3 at day 14 (Fig G). Representing UUO HE and IHC images at day 7 and day 14 are not matching with the scoring data (Figs, CD EF, GH and KL). In some cases, pathological changes (tubular dilatation) are greater in sham group than other groups (Fig C). Inclusion of low magnification of HE and IHC images, complementary protein (Western blot) and mRNA level of fibrosis markers will improve the quality of the manuscript.
Ans: As kindly suggested, we re-evaluated renal pathology with the scoring of HE and IHC changes, and revised the manuscript as follows:
1. Added revised figures (Figure 2D, F, L) to the revised version.
2. The photo of Masson staining for sham control at day 14 was replaced and the revised one is much clearer (Figure 2C of the revised version), to avoid misunderstanding of the pathological changes.
3. It is a very good suggestion to include low magnification of HE and IHC images, complementary protein (Western blot) and mRNA level of fibrosis markers; however, this might involve an extensive and requirement of repeat some experiments in vivo. Therefore, we thank the reviewer’s frank comments very much. As you can see from our point-to-point responses to the comments and suggestions made by the other reviewers (Reviewers #1, #2 and #4) we have made very specific amendment of the manuscript, which might have greatly improved the quality of the revised manuscript.
Point 3: What are the numbers are in scoring graphs in figure 2 and 3B? Are indicating area or mice? Not clear.
Ans: We thank the reviewer very much for the critical comments and suggestions. We added “….Seven mice per group was used.” to the Materials and Methods of the revised version. (Line 137, Page 6).
Point 4: Figures 4, 5 and 6 are representing from how many separate experiments..... are also not indicated in the manuscript.
Ans: We thank the reviewer for the very important comments. Figures 4, 5 and 6 are representing three separate experiments, and each experiment was performed in triplicate. Accordingly, we added the sentence “Data are shown as means ± SEM for three separate experiments, and each experiment was performed in triplicate.” to the Figure Legends of the revised version. (Line 759-760, Page 29; Line 7790-780, Page 30; Line 795-798, Pages 30-31).
Point 5: Is there any effect of NSC828779 on total p38, JNK and ERK?
Ans: We thank the reviewer for the very important comments. Although we did not detect the total form of p38, JNK and ERK, we evaluated the inhibitory effects of NSC828779 on phosphorylation of those kinases by using specific inhibitors, respectively (Figure 5E). Therefore, we added a statement of “In the present study, we evaluated the inhibitory effects of NSC828779 on phosphorylation of p38, JNK and ERK by using specific inhibitors, respectively,although we did not detect the total form of those kinases (Figure 5F). However, this issue is worth further investigation for the mechanism of action of the small molecule.” to the Discussion of the revised version (Line 430-434, Page 17).

Reviewer 4 Report

The authors conducted an interesting study investigating the therapeutic effects of NSC828779 on renal fibrosis in mice. The results are remarkable and the manuscript well written. I have only some comments: 

  1. Why did the authors choose female mice and not mice of both sexes? 
  2. It would be more appropriate to avoid inclusion of results from previous studies in the Results section of the manuscript. The authors may explain their preliminary results in Methodology section. 
  3. How did the authors conclude concerning the dose of drug to administer in mice? Where there any preliminary results regarding the optimal effective dose? 
  4. The authors compare the results between UUO+ vehicle mice and those treated with NSC828779. No comparison was performed between sham controls and the mice treated with NSC828779. This statistical comparison is important in order to understand to which extent this drug is efficient in improving renal fibrosis. Therefore, all tests should include a comparison between sham controls and those treated with the new drug and the results need to be commented in the Discussion section. 
  5. Moreover it is important to compare all the results between 7 and 14 days in mice treated with the new drug, in order to understand the duration of the effects of the new drug.

Author Response

Reviewer #4:
The authors conducted an interesting study investigating the therapeutic effects of NSC828779 on renal fibrosis in mice. The results are remarkable and the manuscript well written. I have only some comments: 
Point 1: Why did the authors choose female mice and not mice of both sexes? 
Ans: We thank the reviewer for the very important comments and suggestions. Both sexes appear to be proper for the study as described previously (FASEB J. 2018 May; 32(5): 2644–2657, Int. J. Mol. Med. 2017;41:95–106.).
References:
Anorga S, Overstreet JM, Falke LL, Tang J, Goldschmeding RG, Higgins PJ, Samarakoon R. Deregulation of Hippo–TAZ pathway during renal injury confers a fibrotic maladaptive phenotype. FASEB J. 2018 May; 32(5): 2644–2657.
Nguyen-Thanh T, Kim D, Lee S, Kim W, Park S, Kang K. Inhibition of histone deacetylase 1 ameliorates renal tubulointerstitial fibrosis via modulation of inflammation and extracellular matrix gene transcription in mice. Int. J. Mol. Med. 2017; 41: 95–106. 

Point 2: It would be more appropriate to avoid inclusion of results from previous studies in the Results section of the manuscript. The authors may explain their preliminary results in Methodology section. 
Ans: We thank the reviewer very much for the critical comments and suggestions. As kindly suggested, we revised it by deleting these statements (Line 286, Page 12; Line 296, Page 12; Line 314, page 13; Line 317, page 13; Line 325, page 13; Line 333, page 13; Line 372, page 15) of the revised version.
Point 3: How did the authors conclude concerning the dose of drug to administer in mice? Where there any preliminary results regarding the optimal effective dose? 
Ans: We thank the reviewer very much for the critical comments and suggestions. The same question was made by Reviewer #1. In our pilot study, the dose used throughout our study, 10mg/kg/day was an optimal dose for the treatment of choice. In compliance with the important comment by the reviewer, we added the following to our revised version: “Besides, the NSC828779 treatment showed no detectable systemic side effects in the mice.” in the Results section of the revised version (Line 255-256, Page 11).
Point 4: The authors compare the results between UUO+ vehicle mice and those treated with NSC828779. No comparison was performed between sham controls and the mice treated with NSC828779. This statistical comparison is important in order to understand to which extent this drug is efficient in improving renal fibrosis. Therefore, all tests should include a comparison between sham controls and those treated with the new drug and the results need to be commented in the Discussion section. 
Ans: We thank the reviewer very much for the critical comments and suggestions.
1. As kindly suggested, we performed statistical comparison for animal models between the sham control and the mice treated with NSC828779 for quantification, added the results to the Results section Figures 1, 2 and 3B of the revised version, and revised Materials and Methods as follows: “Sham-operated mice, which received an identical surgical procedure, but without ureteric ligation (sham control), and sham-operated mice treated with NSC828779 (sham+NSC828779) served as controls.” (Line 136-137; Page 6). Accordingly, we added the sentence “The horizonal dashed line indicates the mean for sham control+NSC828779.” to the Figure Legends of the revised version. (Line 713-714, Page 27; Line 724-725, Page 27; Line 740-741, Page 28).
2. As kindly suggested, we added the issue to the Discussion section of the revised version as follows: “In the present study, a comparison between sham controls and those treated with the compound were made and there were no detectable systemic side effects in the mice that received NSC828779, also justifying it as highly potential drug candidate.” (Line 456-458, Page 18 of the revised version).
Point 5: Moreover it is important to compare all the results between 7 and 14 days in mice treated with the new drug, in order to understand the duration of the effects of the new drug.
Ans: We thank the reviewer very much for the critical comments and suggestions. As kindly suggested, we re-evaluated results between 7 and 14 days in mice with NSC828779 for quantitative analysis and revised it (Figures 1, 2, 3 and 7). Accordingly, we revised certain sentences as follows: “however, there were higher levels of fibrotic changesat day 14 than those of day 7.” (Line 262-263, Page 11); “however, there were higher levels of fibrotic changes at day 14 than those of day 7.’’ (Line 270-272, Page 11); “however, there were higher levels of Col-III at day 14 than those of day 7.’’ (Line 284-285, Page 12); “however, there were higher mRNA levels of NLRP3, IL-1β, and caspase-1 at day 14 than those of day 7” (Line 298-300, Page 12); “however, there was a higher protein level of IL-1β at day 14 than those of day 7.” (Line 311-312, Page 13); “there was a higher level of p-STAT3 at day 14 than those of day 7.” (Line 375, Page 15) to the Results section of the revised version.

Round 2

Reviewer 3 Report

The manuscript has been improved after revision.

Reviewer 4 Report

Thank you. All me major and minor points of review have been answered

I have no further comments